# Use of machine learning to identify risk factors for coronary artery disease

**Alexander A. Huang**[1,2☯*], **Samuel Y. Huang**[1,3☯]

**1** Department of Statistics and Data Science, Cornell University, Ithaca, New York, United States of America, **2** Department of MD Education, Northwestern University Feinberg School of Medicine, Chicago, Illinois, United States of America, **3** Department of Internal Medicine, Virginia Commonwealth University School of Medicine, Richmond, Virginia, United States of America

☯ These authors contributed equally to this work.
* alexander.huang@northwestern.edu

**Data Availability Statement:** The data is freely available without restriction from and can be found on the NHANES section of the CDC website[29]. Data Share Statement: Data described in the manuscript are present at: https://wwwn.cdc.gov/

## Abstract

Coronary artery disease (CAD) is the leading cause of death in both developed and developing nations. The objective of this study was to identify risk factors for coronary artery disease through machine-learning and assess this methodology. A retrospective, cross-sectional cohort study using the publicly available National Health and Nutrition Examination Survey (NHANES) was conducted in patients who completed the demographic, dietary, exercise, and mental health questionnaire and had laboratory and physical exam data. Univariate logistic models, with CAD as the outcome, were used to identify covariates that were associated with CAD. Covariates that had a p<0.0001 on univariate analysis were included within the final machine-learning model. The machine learning model XGBoost was used due to its prevalence within the literature as well as its increased predictive accuracy in healthcare prediction. Model covariates were ranked according to the Cover statistic to identify risk factors for CAD. Shapely Additive Explanations (SHAP) explanations were utilized to visualize the relationship between these potential risk factors and CAD. Of the 7,929 patients that met the inclusion criteria in this study, 4,055 (51%) were female, 2,874 (49%) were male. The mean age was 49.2 (SD = 18.4), with 2,885 (36%) White patients, 2,144 (27%) Black patients, 1,639 (21%) Hispanic patients, and 1,261 (16%) patients of other race. A total of 338 (4.5%) of patients had coronary artery disease. These were fitted into the XGBoost model and an AUROC = 0.89, Sensitivity = 0.85, Specificity = 0.87 were observed (Fig 1). The top four highest ranked features by cover, a measure of the percentage contribution of the covariate to the overall model prediction, were age (Cover = 21.1%), Platelet count (Cover = 5.1%), family history of heart disease (Cover = 4.8%), and Total Cholesterol (Cover = 4.1%). Machine learning models can effectively predict coronary artery disease using demographic, laboratory, physical exam, and lifestyle covariates and identify key risk factors.

## Introduction

Coronary artery disease (CAD) is the leading cause of death in both developed and developing nations [1]. CAD is an atherosclerotic disease that is associated with major complications,

nchs/nhanes/continuousnhanes/default.aspx?cycle=2017-2020. Analytic code is present at: https://github.com/samuelyhuang/Machine-Learning-to-Identify-Risk-Factors-for-Coronary-Artery-Disease.

**Funding:** The authors received no specific funding for this work.

**Competing interests:** The authors have declared that no competing interests exist.

including angina, myocardial infarction, and sudden cardiac death [2–5]. Due to the high prevalence, morbidity, and mortality of CAD, identification of risk factors is a public health priority [6]. Genome-wide association studies have identified several genetic variants linked to CAD [7–10]. Additionally, epidemiological studies have identified significant socioeconomic, race, and sex disparities in CAD prevalence, quality measures, and outcomes [1, 11–13]. Further work has found that a combination of genetic, demographic, and environmental factors contributes to the severity of CAD and other cardiovascular diseases [1, 14–17]. Furthermore, lifestyle factors, such as diet and exercise, have been found to play an important role in the risk for CAD and other cardiovascular diseases [6, 18–20]. These studies have been combined to develop joint risk scores, factoring in both physiological covariates (blood pressure, cholesterol) as well as demographic covariates (age, race, gender) [5, 8, 9, 21]. Despite the strong literature studying the risk factors for CAD, most studies focus upon hypothesis testing or epidemiology focusing upon specific risk factors of interest [22–24]. While CAD is recognized as being of "multifactorial" cause, little is known regarding the relative predictive power of different risk factors (lifestyle vs genetic vs chronic disease comorbidities). Given these limitations in the literature, we will leverage transparent machine-learning methods including Shapely Additive Explanations (SHAP model explanations) and model gain statistics to identify pertinent risk-factors for CAD and compute their relative contribution to model prediction of CAD risk; the NHANES 2017–2020 cohort, a large, nationally representative sample of US adults, will be used within this study.

## Methods

A retrospective, cross-sectional cohort study using the publicly available National Health and Nutrition Examination Survey (NHANES) was conducted in patients who completed the demographic, dietary, exercise, and mental health questionnaire and had laboratory and physical exam data.

### Ethics approval and consent to participate

The acquisition and analysis of the data within this study was approved by the National Center for Health Statistics Ethics Review Board.

### Dataset and cohort selection

The National Health and Nutrition Examination Survey (NHANES 2017–2020) is a program designed by the National Center for Health Statistics (NCHS), which has been leveraged to assess the health and nutritional status of the United States population [25]. The NHANES dataset is a series of cross-sectional, complex, multi-stage surveys conducted by the Centers for Disease Control and Prevention (CDC) on a nationally representative cohort of the United States population to provide health, nutritional, and physical activity data. In the present study, we analyzed adult ($\geq$18 years old) patients in the NHANES dataset who completed the demographic, dietary, exercise, and mental health questionnaire and had laboratory and physical exam data.

### Assessment of coronary artery disease

The medical conditions file was used to define coronary artery disease. Participants were asked: "Has a doctor or other health professional ever told you that you have coronary heart disease?" Participants who answered "Yes" to this question were considered as having CAD within this study.

### Independent variable

Potential model covariates were identified within the demographics, dietary, physical examination, laboratory, and medical questionnaire datasets in NHANES. All covariates were extracted and merged with the CAD indicator.

### Model construction and statistical analysis

Univariate logistic models, with CAD as the outcome, were used to identify covariates that were associated with CAD. Covariates that had a p<0.0001 on univariate analysis were included within the final machine-learning model. The machine learning model XGBoost was used due to its prevalence within the literature as well as its increased predictive accuracy in healthcare prediction. XGBoost models were fit with a train:test (80:20), and model accuracy statistics (AUROC, Sensitivity, Specificity, F1, Balanced Accuracy) were computed. Model covariates were ranked according to the Gain, Cover, and Frequency (representations of the relative contribution ("model importance") of each of the covariates) to identify risk factors for CAD. The Gain statistic represents the overall proportion of the model prediction is attributed to a given statistic. The Cover and Frequency are representations of the proportion of trees that each of the covariates appear within the machine-learning model. SHAP explanations were utilized to visualize the relationship between these potential risk factors and CAD.

## Results

Table 1 shows that f the 7,929 patients that met the inclusion criteria in this study, 4,055 (51%) were female, 2,874 (49%) were male. The mean age was 49.2 (SD = 18.4), with 2,885 (36%) White patients, 2,144 (27%) Black patients, 1,639 (21%) Hispanic patients, and 1,261 (16%) patients of another race. A total of 338 (4.5%) of patients had coronary artery disease.

The machine learning model had 58 out of a total 684 features that were found to be significant on univariate analysis (P<0.0001 used). These were fitted into the XGBoost model and an AUROC = 0.89, Sensitivity = 0.85, Specificity = 0.87 were observed Fig 1.

Table 2 shows the top five highest ranked features by cover, a measure of the percentage contribution of the covariate to the overall model prediction, were age (Cover = 21.1%), Platelet count (Cover = 5.1%), family history of heart disease (Cover = 4.8%), and Total Cholesterol (Cover = 4.1%).

In Fig 2, on SHAP visualization, we observed that: interpret the top four covariates age had a sigmoidal relationship with risk for coronary artery disease.

Figs 3, 4a and 4b shows the SHAP Explanations for various SHAP features. We observed that at ages between 20 and 35, there was no significant change in risk for CAD with increasing age, with age increasing between 35 and 70, there was a significant increase in risk for CAD with increasing age, and above 70 years of age, there was no significant increase in CAD with increasing age. Additionally, a curvilinear relationship was observed analyzing the relationship with total-cholesterol and risk for CAD. Patients with significantly decreased total cholesterol were observed to have increased risk for heart disease, and patients with increased cholesterol were observed to also be at increased risk, with a minimum risk around 200 mg/dL of cholesterol. A curvilinear relationship was also observed for the relationship between platelet count and risk for CAD, with significantly decreased platelet counts linked with CAD and significantly increased platelet counts also linked with CAD, a minimum observed around 300,000 cells/uL. Family history was also a significant predictor for CAD. Patients with close relatives having a heart attack in the past had significant increased risk for CAD.

**Table 1. Demographic information.**

| Coronary Artery Disease Grouping | All Patients | Coronary Heart Disease | No Coronary Heart Disease | P-Values |
|---|---|---|---|---|
| Age; Mean (SD) | 49.23 (18.35) | 68.34 (10.49) | 48.38 (18.16) | P<0.001 |
| Gender Female; Count (Proportion) | 4055 (0.51) | 92 (0.27) | 3963 (0.52) | 0.19 |
| Gender Male; Count (Proportion) | 3874 (0.49) | 246 (0.73) | 3628 (0.48) | |
| Race White; Count (Proportion) | 2885 (0.36) | 210 (0.62) | 2675 (0.35) | 0.21 |
| Race Black; Count (Proportion) | 2144 (0.27) | 47 (0.14) | 2097 (0.28) | |
| Race Hispanic; Count (Proportion) | 1639 (0.21) | 48 (0.14) | 1591 (0.21) | |
| Race Other; Count (Proportion) | 1261 (0.16) | 33 (0.1) | 1228 (0.16) | |
| Income_Poverty_Ratio; Mean (SD) | 2.6 (1.63) | 2.68 (1.53) | 2.6 (1.64) | P<0.001 |
| Albumin, urine (mg/L); Mean (SD) | 47.96 (272.25) | 159 (695.36) | 43.02 (235.41) | P<0.001 |
| Creatinine, urine (mg/dL); Mean (SD) | 132.8 (87.58) | 115.54 (69.81) | 133.56 (88.21) | P<0.001 |
| Albumin creatinine ratio (mg/g); Mean (SD) | 46.65 (296.14) | 167.66 (773.04) | 41.28 (253.85) | P<0.001 |
| Direct HDL-Cholesterol (mg/dL); Mean (SD) | 53.51 (16.03) | 48.27 (13.22) | 53.75 (16.1) | P<0.001 |
| LDL-Cholesterol, Friedewald (mg/dL); Mean (SD) | 107.67 (35.57) | 88.52 (38.53) | 108.53 (35.2) | P<0.001 |
| Total Cholesterol (mg/dL); Mean (SD) | 184.54 (41.08) | 163.6 (42.63) | 185.49 (40.76) | P<0.001 |
| Lymphocyte percent (%); Mean (SD) | 31.28 (8.96) | 26.97 (9.08) | 31.47 (8.9) | P<0.001 |
| Monocyte percent (%); Mean (SD) | 8.19 (2.23) | 8.77 (2.3) | 8.17 (2.22) | P<0.001 |
| Segmented neutrophils percent (%); Mean (SD) | 57.07 (9.66) | 60.4 (9.42) | 56.92 (9.65) | P<0.001 |
| Eosinophils percent (%); Mean (SD) | 2.78 (2.08) | 3.16 (2.11) | 2.77 (2.08) | P<0.001 |
| Monocyte number (1000 cells/uL); Mean (SD) | 0.58 (0.21) | 0.63 (0.2) | 0.57 (0.21) | P<0.001 |
| Segmented neutrophils num (1000 cell/uL); Mean (SD) | 4.18 (1.71) | 4.5 (1.54) | 4.17 (1.72) | P<0.001 |
| Eosinophils number (1000 cells/uL); Mean (SD) | 0.2 (0.17) | 0.23 (0.16) | 0.2 (0.17) | P<0.001 |
| Mean cell volume (fL); Mean (SD) | 88.4 (6.19) | 89.93 (5.94) | 88.33 (6.19) | P<0.001 |
| Red cell distribution width (%); Mean (SD) | 13.9 (1.39) | 14.33 (1.59) | 13.88 (1.37) | P<0.001 |
| Platelet count (1000 cells/uL); Mean (SD) | 246.35 (65.54) | 212.2 (58.43) | 247.87 (65.43) | P<0.001 |
| RBC folate (ng/mL); Mean (SD) | 514.51 (230.32) | 676.1 (345.28) | 509.04 (223.43) | P<0.001 |
| Serum total folate (nmol/L); Mean (SD) | 39.29 (27.09) | 58.23 (46.94) | 38.64 (25.89) | P<0.001 |
| 5-Methyl-tetrahydrofolate (nmol/L); Mean (SD) | 36.46 (22.3) | 53.15 (41.16) | 35.88 (21.12) | P<0.001 |
| Tetrahydrofolate (nmol/L); Mean (SD) | 0.77 (0.57) | 1.08 (0.8) | 0.76 (0.56) | P<0.001 |
| Mefox oxidation product (nmol/L); Mean (SD) | 1.75 (1.91) | 2.81 (2.27) | 1.71 (1.88) | P<0.001 |
| Glycohemoglobin (%); Mean (SD) | 5.82 (1.09) | 6.42 (1.32) | 5.79 (1.08) | P<0.001 |
| Blood lead (ug/dL); Mean (SD) | 1.16 (1.16) | 1.57 (1.31) | 1.14 (1.15) | P<0.001 |
| Fasting Glucose (mg/dL); Mean (SD) | 112.57 (36.98) | 131.06 (48.49) | 111.75 (36.18) | P<0.001 |
| Alkaline Phosphatase (ALP) (IU/L); Mean (SD) | 78.23 (26.94) | 85.84 (30.2) | 77.89 (26.73) | P<0.001 |
| Blood Urea Nitrogen (mg/dL); Mean (SD) | 14.86 (6.02) | 20.06 (9.06) | 14.63 (5.74) | P<0.001 |
| Creatinine, refrigerated serum (mg/dL); Mean (SD) | 0.91 (0.5) | 1.15 (0.75) | 0.89 (0.48) | P<0.001 |
| Glucose, refrigerated serum (mg/dL); Mean (SD) | 101.39 (35.15) | 116.48 (42.96) | 100.7 (34.6) | P<0.001 |
| Lactate Dehydrogenase (LDH) (IU/L); Mean (SD) | 158.13 (35.03) | 166.62 (39.66) | 157.74 (34.76) | P<0.001 |
| Osmolality (mmol/Kg); Mean (SD) | 281.34 (5.61) | 284.06 (7.03) | 281.21 (5.5) | P<0.001 |
| Potassium (mmol/L); Mean (SD) | 4.09 (0.36) | 4.27 (0.43) | 4.08 (0.35) | P<0.001 |
| Total Bilirubin (mg/dL); Mean (SD) | 0.46 (0.28) | 0.52 (0.3) | 0.45 (0.28) | P<0.001 |
| Cholesterol, refrigerated serum (mg/dL); Mean (SD) | 184.87 (41.12) | 164 (42.45) | 185.81 (40.8) | P<0.001 |
| Total Protein (g/dL); Mean (SD) | 7.15 (0.45) | 7.02 (0.46) | 7.16 (0.45) | P<0.001 |
| Uric acid (mg/dL); Mean (SD) | 5.4 (1.47) | 5.93 (1.5) | 5.37 (1.47) | P<0.001 |
| BMXWT—Weight (kg); Mean (SD) | 84.02 (23.31) | 88.16 (22.7) | 83.83 (23.33) | P<0.001 |
| BMXWAIST—Waist Circumference (cm); Mean (SD) | 100.65 (17.47) | 108.3 (14.99) | 100.33 (17.5) | P<0.001 |
| SMQ020—Smoked at least 100 cigarettes in life; Mean (SD) | 3261 (0.41) | 208 (0.62) | 3053 (0.4) | P<0.001 |
| SMQ681—Smoked tobacco last 5 days? 1; Mean (SD) | 1722 (0.22) | 49 (0.14) | 1673 (0.22) | P<0.001 |

*(Continued)*

**Table 1.** (Continued)

| Coronary Artery Disease Grouping | All Patients | Coronary Heart Disease | No Coronary Heart Disease | P-Values |
|---|---|---|---|---|
| MCQ053—Taking treatment for anemia/past 3 mos 1; Mean (SD) | 412 (0.05) | 33 (0.1) | 379 (0.05) | P<0.001 |
| MCQ080—Doctor ever said you were overweight 1; Mean (SD) | 3175 (0.4) | 187 (0.55) | 2988 (0.39) | P<0.001 |
| MCQ092—Ever receive blood transfusion 1; Mean (SD) | 879 (0.11) | 90 (0.27) | 789 (0.1) | P<0.001 |
| MCQ160a—Doctor ever said you had arthritis 2; Mean (SD) | 5188 (0.65) | 136 (0.4) | 5052 (0.67) | P<0.001 |
| MCQ160a—Doctor ever said you had arthritis 1; Mean (SD) | 2345 (0.3) | 202 (0.6) | 2143 (0.28) | P<0.001 |
| MCQ160m—Ever told you had thyroid problem 1; Mean (SD) | 908 (0.11) | 77 (0.23) | 831 (0.11) | P<0.001 |
| MCQ160p—Ever told you had COPD, emphysema, ChB 1; Mean (SD) | 717 (0.09) | 101 (0.3) | 616 (0.08) | P<0.001 |
| MCQ520—Abdominal pain during past 12 months? 1; Mean (SD) | 1693 (0.21) | 120 (0.36) | 1573 (0.21) | P<0.001 |
| MCQ300c—Close relative had diabetes? 1; Mean (SD) | 3653 (0.46) | 211 (0.62) | 3442 (0.45) | P<0.001 |
| MCQ300a—Close relative had heart attack? 1; Mean (SD) | 1011 (0.13) | 109 (0.32) | 902 (0.12) | P<0.001 |
| MCQ366a—Doctor told you to control/lose weight 1; Mean (SD) | 2275 (0.29) | 151 (0.45) | 2124 (0.28) | P<0.001 |
| PHQ_9; Mean (SD) | 3.18 (4.3) | 4.2 (5.75) | 3.13 (4.22) | P<0.001 |
| Caffeine..mg..1; Mean (SD) | 137.22 (202.64) | 183.64 (269.28) | 135.16 (198.93) | P<0.001 |
| Median.liver.stiffness; Mean (SD) | 6.05 (5.26) | 7.52 (8.3) | 5.99 (5.09) | P<0.001 |
| Direct.HDL.Cholesterol..mg.dL.; Mean (SD) | 53.51 (16.03) | 48.27 (13.22) | 53.75 (16.1) | P<0.001 |
| LBDLDL. . .LDL.Cholesterol..Friedewald..mg.dL.; Mean (SD) | 107.67 (35.57) | 88.52 (38.53) | 108.53 (35.2) | P<0.001 |
| Systolic_1; Mean (SD) | 124.09 (19.51) | 131.52 (23.28) | 123.76 (19.26) | P<0.001 |
| Diastolic_1; Mean (SD) | 74.85 (11.76) | 71.46 (12.71) | 75 (11.7) | P<0.001 |

Descriptive statistics for demographic characteristics and all covariates within the machine learning model, stratified by whether patients had coronary artery disease.

## Discussion

In this retrospective, cross sectional cohort of United States adults, a machine learning model utilizing demographic, laboratory, physical examination, and lifestyle questionnaire data had

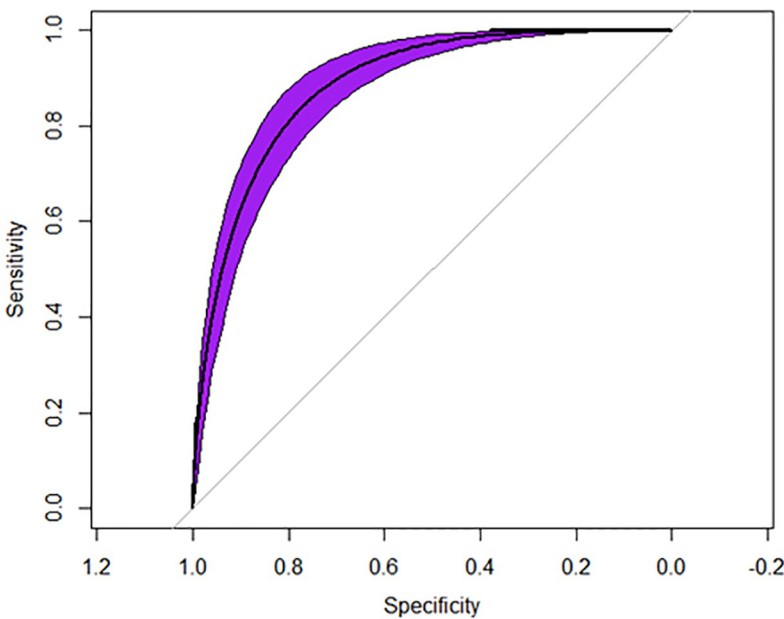

**Fig 1. Restricted operator characteristic curve and model statistics.** The ROC Curve for the machine-learning model predicting coronary artery disease. AUROC = 0.89.

**Table 2. Model feature importance statistics.**

| Feature | Gain | Cover | Frequency |
|---|---|---|---|
| Age | 19.8% | 22.0% | 5.5% |
| Platelet count (1000 cells/uL) | 4.7% | 3.7% | 4.3% |
| Albumin creatinine ratio (mg/g) | 3.9% | 4.8% | 3.9% |
| Total Cholesterol (mg/dL) | 3.6% | 3.4% | 3.2% |
| MCQ300a—Close relative had heart attack? | 3.4% | 6.3% | 2.7% |
| Median.liver.stiffness | 3.0% | 3.5% | 3.9% |
| Diastolic_1 | 2.9% | 2.0% | 3.4% |
| Blood Urea Nitrogen (mg/dL) | 2.5% | 3.9% | 2.1% |
| Caffeine..mg..1 | 2.3% | 2.0% | 2.9% |
| Blood lead (ug/dL) | 2.3% | 1.7% | 3.2% |
| Segmented neutrophils num (1000 cell/uL) | 2.3% | 1.5% | 2.3% |
| Â Potassium (mmol/L) | 2.2% | 2.6% | 3.0% |
| Cholesterol, refrigerated serum (mg/dL) | 2.1% | 1.7% | 1.7% |
| Gender | 2.0% | 2.1% | 1.3% |
| Â Direct HDL-Cholesterol (mg/dL) | 1.8% | 1.3% | 2.0% |
| Lymphocyte percent (%) | 1.8% | 1.3% | 2.1% |
| MCQ160p—Ever told you had COPD, emphysema, ChB | 1.8% | 1.4% | 1.2% |
| BMXWT—Weight (kg) | 1.8% | 3.4% | 3.4% |
| Systolic_1 | 1.7% | 1.0% | 2.4% |
| Mefox oxidation product (nmol/L) | 1.6% | 4.3% | 2.8% |
| BMXWAIST—Waist Circumference (cm) | 1.6% | 1.6% | 2.3% |
| Albumin, urine (mg/L) | 1.6% | 2.0% | 2.5% |
| PHQ_9 | 1.6% | 1.6% | 1.7% |
| Eosinophils percent (%) | 1.6% | 0.8% | 1.9% |
| Â Glycohemoglobin (%) | 1.6% | 2.2% | 1.7% |
| Segmented neutrophils percent (%) | 1.5% | 0.8% | 1.7% |
| Creatinine, urine (mg/dL) | 1.5% | 0.5% | 2.1% |
| Â Alkaline Phosphatase (ALP) (IU/L) | 1.5% | 1.0% | 2.1% |
| Lactate Dehydrogenase (LDH) (IU/L) | 1.5% | 0.4% | 2.3% |
| LDL-Cholesterol, Friedewald (mg/dL) | 1.4% | 1.7% | 1.9% |
| Red cell distribution width (%) | 1.3% | 1.0% | 1.8% |
| Creatinine, refrigerated serum (mg/dL) | 1.1% | 2.5% | 1.6% |
| Osmolality (mmol/Kg) | 1.0% | 0.9% | 1.3% |
| Uric acid (mg/dL) | 1.0% | 0.4% | 1.3% |
| RBC folate (ng/mL) | 1.0% | 0.9% | 1.5% |
| Â Serum total folate (nmol/L) | 0.9% | 0.4% | 1.0% |
| Tetrahydrofolate (nmol/L) | 0.8% | 0.3% | 1.2% |
| MCQ520—Abdominal pain during past 12 months? | 0.8% | 0.6% | 0.7% |
| Â Glucose, refrigerated serum (mg/dL) | 0.8% | 0.4% | 1.2% |
| Mean cell volume (fL) | 0.8% | 0.8% | 1.8% |
| Total Protein (g/dL) | 0.8% | 1.3% | 1.2% |
| MCQ092—Ever receive blood transfusion | 0.7% | 0.6% | 0.7% |
| Monocyte percent (%) | 0.7% | 0.2% | 1.1% |
| MCQ300c—Close relative had diabetes? | 0.7% | 0.4% | 0.5% |
| 5-Methyl-tetrahydrofolate (nmol/L) | 0.6% | 0.3% | 0.7% |
| Fasting Glucose (mg/dL) | 0.6% | 0.2% | 0.9% |
| Monocyte number (1000 cells/uL) | 0.6% | 0.2% | 0.6% |

*(Continued)*

**Table 2.** (Continued)

| Feature | Gain | Cover | Frequency |
|---|---|---|---|
| Â Total Bilirubin (mg/dL) | 0.6% | 0.3% | 0.9% |
| MCQ366a—Doctor told you to control/lose weight | 0.5% | 0.6% | 0.5% |
| MCQ080—Doctor ever said you were overweight | 0.5% | 0.4% | 0.4% |
| SMQ020—Smoked at least 100 cigarettes in life | 0.2% | 0.2% | 0.3% |
| MCQ160a—Doctor ever said you had arthritis | 0.2% | 0.2% | 0.3% |
| MCQ160m—Ever told you had thyroid problem | 0.2% | 0.1% | 0.3% |
| SMQ681—Smoked tobacco last 5 days? | 0.1% | 0.2% | 0.4% |
| Eosinophils number (1000 cells/uL) | 0.1% | 0.2% | 0.2% |
| MCQ053—Taking treatment for anemia/past 3 mos | 0.1% | 0.1% | 0.1% |

The Cover, Gain, and Frequency of all covariates within the XGBoost model.

strong predictive accuracy (AUROC = 0.89). The greatest predictors for coronary artery disease included age, total cholesterol, total platelets, and family history of a heart attack.

The visualizations completed for the top four covariates were concordant with current literature around the relationship between these covariates and coronary artery disease: there is strong epidemiological and physiological evidence for the link between increased age and cholesterol as major risk factors for coronary artery disease [26–28]. The non-linear relationship between cholesterol and coronary artery disease matches survival-modeling and restricted

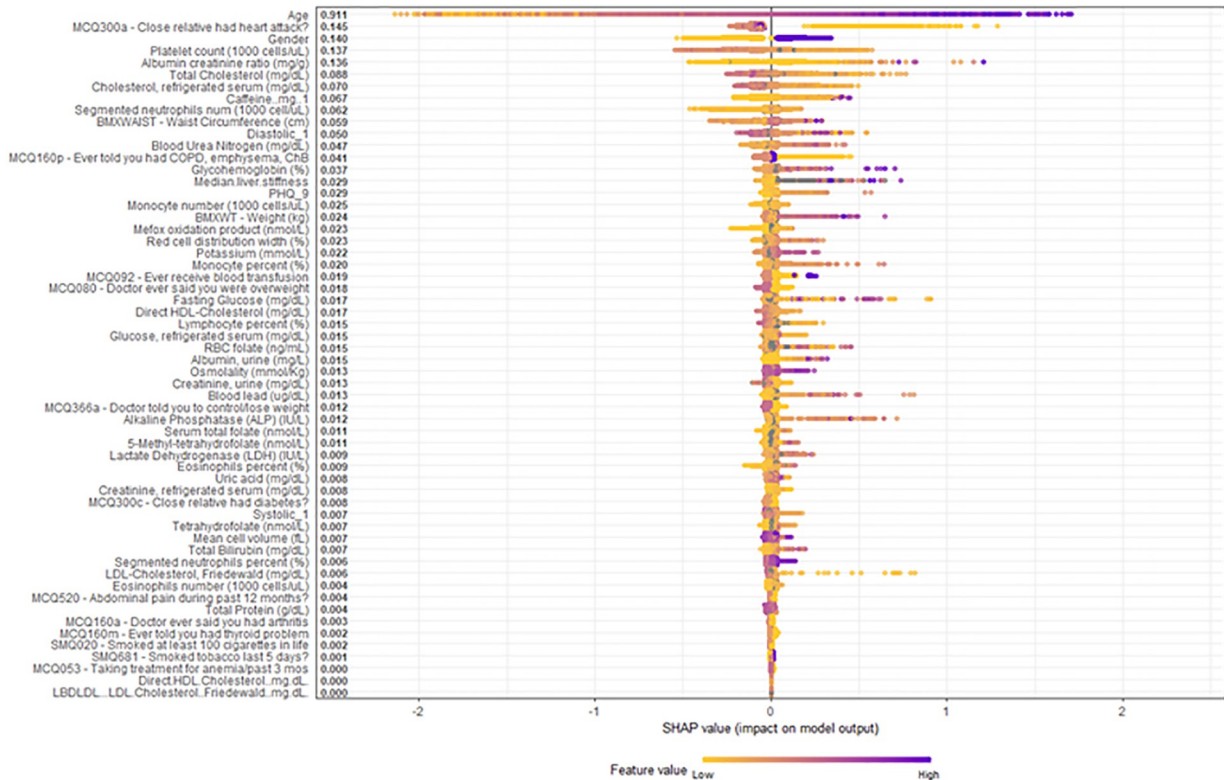

**Fig 2. Overall SHAP explanations.** SHAP explanations, purple color representing higher values of the covariate while yellow representing lower values of the covariate. X-axis is the change in log-odds for CAD.

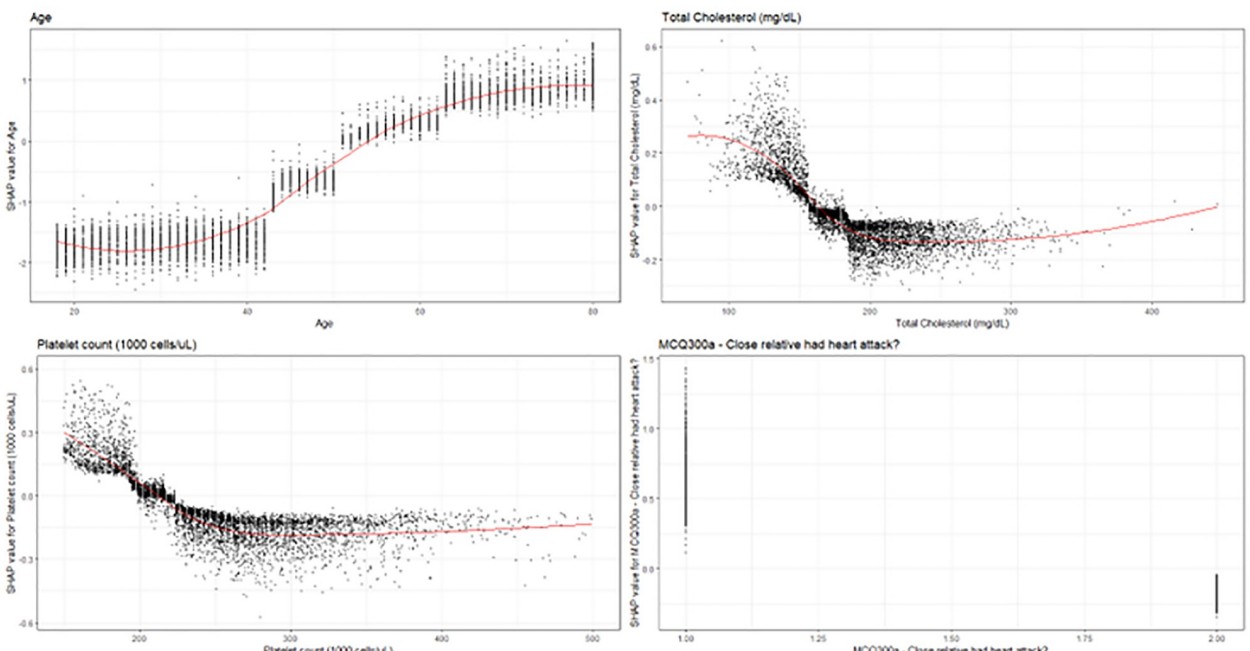

**Fig 3. SHAP explanations for the top 4 covariates.** SHAP explanations, covariate value on the x-axis, change in log-odds on the y-axis, red line represents the relationship between the covariate and log-odds for CAD, each black dot represents an observation. Covariates (top left—Age, top right—Total Cholesterol, bottom left—platelets, bottom right—Close relative with a heart attack (Yes = 1, no = 2)).

cubic spline analysis from other studies [28–36]. Furthermore, multiple genetic and sociological studies have found that family history is a significant risk factor for coronary artery disease [37–40]. Additionally, low-platelets being associated with coronary artery disease is associated with pathology such as thrombocytopenia [41–43]. In addition to the top four covariates within our model, we also wanted to explore if the machine-learning model was able to generate predictions for HDL-Cholesterol and Systolic Blood pressure, two major risk factors that have been widely studied within the cardiovascular literature. In these visualizations, we observed a strong negative relationship between HDL-cholesterol and risk for coronary artery disease (Fig 4a). We observe a curvilinear relationship between systolic blood pressure and coronary heart disease, with blood pressures lower than 120 being associated with increased risk for coronary heart disease and blood pressures above being strongly associated with coronary heart disease as well.

Since visualizations for risk factors match literature relationships, we have increased confidence that the machine learning model is able to capture the actual physiological relationships of these covariates [44–46]. These transparent machine-learning tools allow for increased confidence that these algorithms are picking up true signal within these covariates to predict coronary artery disease rather than just replicating potential biases stemming from systemic data-= quality errors that are present within the dataset. Additionally, these SHAP visualizations allow us to interpret that the increase predictive power of these machine-learning methods is associated with the ability for these non-parametric methods to more accurately capture the non-linear interactive relationship between the covariates, rather than just over-fitting the model to get increased accuracy.

The greatest strength of this algorithmic method for identification of the covariates is the ability to search through hundreds of covariates systematically without relying upon judgment form the researcher, which may be muddled by potential personal biases. This method also

(A)

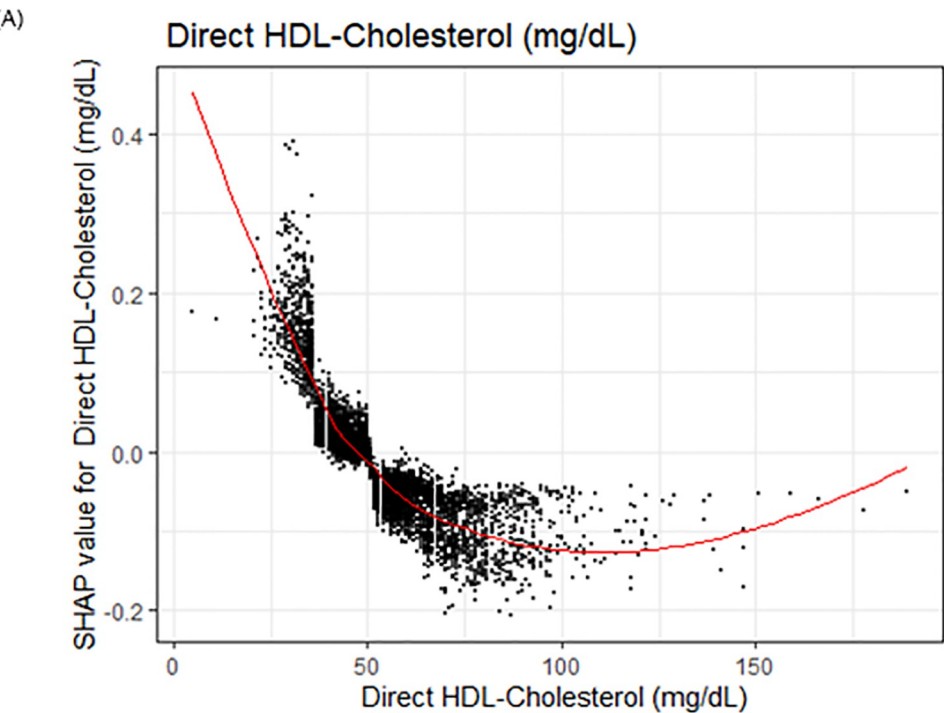

(B)

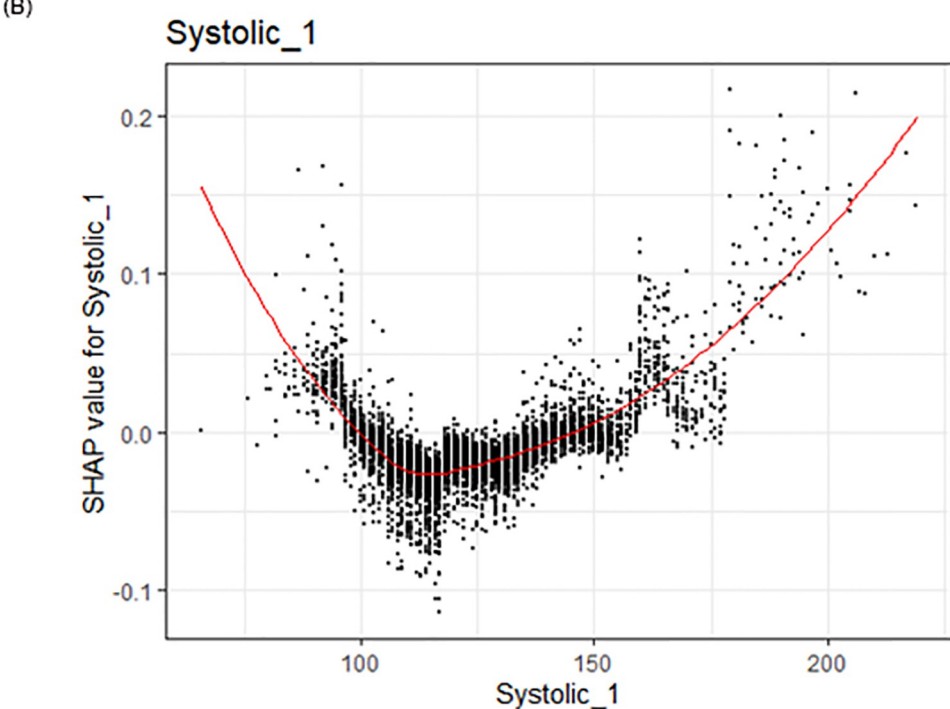

**Fig 4. a**: Covariates of interest to evaluate sensibility of the model. SHAP explanations for the relationship between HDL-Cholesterol and odds of CAD. Covariate value on the x-axis, change in log-odds on the y-axis, red line represents the relationship between the covariate and log-odds for CAD, each black dot represents an observation. **b**: SHAP explanations for the relationship between Systolic blood pressure and odds of CAD. Covariate value on the x-axis, change in log-odds on the y-axis, red line represents the relationship between the covariate and log-odds for CAD, each black dot represents an observation.

allows for the ranking of the relative importance of each of these covariates through the cover statistic, which allows us to obtain the relative contribution to the prediction each covariate has and thus infer from there an estimate for the relative contribution to true risk for coronary artery disease that each patient has. Another strength is that after these covariates are selected and the model built, SHAP visualizations can be used to make sure that each of the covariate either matches current literature understandings of the covariate's association with coronary heart disease or in the case of a discrepancy, allow researchers to validate the plausibility of this feature and then evaluate for potential errors in data-quality.

Some potential weaknesses to this machine-learning analysis is the necessity of the retrospective nature of this cohort. The covariates that were selected within this study will be better at predicting coronary heart disease risk for this cohort than for other cohorts. However, this was limited by the use of training: testing sets to be able to minimize the errors that come with overfitting. Furthermore, visualizations of SHAP allow researchers to test for physiologic plausibility of each of these covariates and allows for effective analysis by researchers of whether these effects are due to true signal or if they are just noise that may be contributing to a type-1 error.

Given the analysis of the strengths and weaknesses of these methods, we argue that use of machine-learning methods can be an effective first step in the identification of risk-factors that can then be further selected by clinicians based upon the specific clinical presentation.

## Limitations

This study has several strengths and weaknesses. We utilized the NHANES dataset, which is a retrospective cohort, carrying the limitations of retrospective studies. However, this study allows for the selection of a large cohort, evaluation of data quality, and due to the publicly available nature of the cohort, allows for increased replication and follow-up studies based upon the same cohort. Furthermore, the cohort relied on surveys to obtain the outcome of interest (CAD) as well as the dietary and lifestyle information. More accurate measurements may have been achieved with prospective studies with automated measurement of foods. However, self-reported survey information allows for the volume of participants to be included within this study. Another weakness was the voluntary nature of this cohort, with participants choosing to opt into the study instead of being randomly selected. This may artificially select a different cohort that may significantly differ from the population. However, our analysis found a demographically diverse population, so these results may still be generalizable to other cohorts.

## Conclusion

Machine learning models can effectively predict coronary artery disease using demographic, laboratory, physical exam, and lifestyle covariates. Age, total cholesterol, total platelets, and family history of heart attack are the strongest predictors of coronary artery disease.

## Acknowledgments

The authors acknowledge the NHANES dataset, which was used in this study.

## Author Contributions

**Conceptualization:** Alexander A. Huang, Samuel Y. Huang.

**Data curation:** Alexander A. Huang, Samuel Y. Huang.

**Formal analysis:** Alexander A. Huang.

**Methodology:** Alexander A. Huang.

**Project administration:** Alexander A. Huang.

**Resources:** Alexander A. Huang.

**Software:** Alexander A. Huang.

**Supervision:** Alexander A. Huang.

**Validation:** Alexander A. Huang.

**Visualization:** Alexander A. Huang.

**Writing – original draft:** Alexander A. Huang, Samuel Y. Huang.

**Writing – review & editing:** Alexander A. Huang, Samuel Y. Huang.

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
