## [Decision Letter · Decision Letter 0]

6 Mar 2023

PONE-D-22-32680Use of Machine Learning to Identify Risk Factors for Coronary Artery DiseasePLOS ONE

Dear Dr. Huang,

Thank you for submitting your manuscript to PLOS ONE. After careful consideration, we feel that it has merit but does not fully meet PLOS ONE’s publication criteria as it currently stands. Therefore, we invite you to submit a revised version of the manuscript that addresses the points raised during the review process.

Please submit your revised manuscript by April 21, 2023. If you will need more time than this to complete your revisions, please reply to this message or contact the journal office at plosone@plos.org. Please include the following items when submitting your revised manuscript:A rebuttal letter that responds to each point raised by the academic editor and reviewer(s). You should upload this letter as a separate file labeled 'Response to Reviewers'.A marked-up copy of your manuscript that highlights changes made to the original version. You should upload this as a separate file labeled 'Revised Manuscript with Track Changes'.An unmarked version of your revised paper without tracked changes. You should upload this as a separate file labeled 'Manuscript'.If applicable, we recommend that you deposit your laboratory protocols in protocols.io to enhance the reproducibility of your results. Protocols.io assigns your protocol its own identifier (DOI) so that it can be cited independently in the future. For instructions see: https://journals.plos.org/plosone/s/submission-guidelines#loc-laboratory-protocols. Additionally, PLOS ONE offers an option for publishing peer-reviewed Lab Protocol articles, which describe protocols hosted on protocols.io. Read more information on sharing protocols at https://plos.org/protocols?utm_medium=editorial-email&utm_source=authorletters&utm_campaign=protocols.

We look forward to receiving your revised manuscript.

Kind regards,

Donovan Anthony McGrowder, PhD., MA., MSc

Academic Editor

PLOS ONE

Journal Requirements:

Additional Editor Comments (if provided):

Dear Dr. Huang,

Your manuscript “Use of Machine Learning to Identify Risk Factors for Coronary Artery Disease” has been assessed by our reviewers. They have raised a number of points which we believe would improve the manuscript and may allow a revised version to be published in PLOS ONE. Their reports, together with any other comments, are below.

 If you are able to fully address these points, we would encourage you to submit a revised manuscript to PLOS ONE.

 Best regards,

Dr. Donovan McGrowder

Reviewers' comments:

Reviewer's Responses to Questions

**Comments to the Author**

1. Is the manuscript technically sound, and do the data support the conclusions?

Reviewer #1: Yes

Reviewer #2: Yes

2. Has the statistical analysis been performed appropriately and rigorously? 

Reviewer #1: Yes

Reviewer #2: Yes

3. Have the authors made all data underlying the findings in their manuscript fully available?

Reviewer #1: Yes

Reviewer #2: Yes

4. Is the manuscript presented in an intelligible fashion and written in standard English?

Reviewer #1: Yes

Reviewer #2: Yes

5. Review Comments to the Author

Reviewer #1: General comments

Integrating the Machine-learning model with the usual statistical analysis model can be a promising revolution in e-Health ad m-Health to solve complex issues accurately and effectively.

The crux of this manuscript is the use of new analytical techniques to complement the usual statistical model. The authors have used machine learning to predict the risk of CAD by capturing the actual physiological relationship of the factors associated with the risk of CAD. The manuscript is well written: methodology is well narrated and executed; the results are well presented, and the discussion is well articulated.

The manuscript lacks page numbers; please insert them in the next version of the manuscript.

Specific comments

Abstract:

- Line 24: The author should define the abbreviation “SHAP.” Most abbreviations must be defined upon first use unless otherwise explained.

Methods:

- Line 91: Why p<0.0001 was the cut-off level of statistical significance to include covariate in the machine-learning model? Is there any reference or statistical explanation?

Discussion:

- Line 200: Misspelled word - Replace “contrary” artery disease with “coronary” artery disease.

Reviewer #2: Thanks very much for this very interesting and informative article. I feel that overall the paper was well written and without significant flaws. The study objectives and the methods section are clearly defined, the article is easily readable, and the topic is relevant to the readership. Limitations are adequately addressed. Conclusions are appropriate for the scope of the study. The paper is formally correct and it is clear its clinical relevance, and what this article should add to the body of knowledge on this topic.

---

## [Author Response · Author response to Decision Letter 0]

13 Mar 2023

Reviewer #1: Yes

Reviewer #2: Yes

2. Has the statistical analysis been performed appropriately and rigorously?

Reviewer #1: Yes

Reviewer #2: Yes

3. Have the authors made all data underlying the findings in their manuscript fully available?

Reviewer #1: Yes

Reviewer #2: Yes

4. Is the manuscript presented in an intelligible fashion and written in standard English?

Reviewer #1: Yes

Reviewer #2: Yes

5. Review Comments to the Author

Reviewer #1: General comments

Integrating the Machine-learning model with the usual statistical analysis model can be a promising revolution in e-Health ad m-Health to solve complex issues accurately and effectively.

The crux of this manuscript is the use of new analytical techniques to complement the usual statistical model. The authors have used machine learning to predict the risk of CAD by capturing the actual physiological relationship of the factors associated with the risk of CAD. The manuscript is well written: methodology is well narrated and executed; the results are well presented, and the discussion is well articulated.

The manuscript lacks page numbers; please insert them in the next version of the manuscript.

###Thank you for your suggestion. The authors have added page numbers.

Specific comments

Abstract:

- Line 24: The author should define the abbreviation “SHAP.” Most abbreviations must be defined upon first use unless otherwise explained.

### Thank you for your suggestion. The authors have defined the abbreviation SHAP on the first instance of its use.

Methods:

- Line 91: Why p<0.0001 was the cut-off level of statistical significance to include covariate in the machine-learning model? Is there any reference or statistical explanation?

##Thank you for your inquiry. A p-value of 0.05 is commonly used. Since we had 684 covariates – we adjusted for multiple testing with a Bonferroni adjustment, the most conservative methodology of adjustment. Then we rounded to the nearest value.

Discussion:

- Line 200: Misspelled word - Replace “contrary” artery disease with “coronary” artery disease.

### Thank you for your suggestion. The authors have made the edit.

Reviewer #2: Thanks very much for this very interesting and informative article. I feel that overall the paper was well written and without significant flaws. The study objectives and the methods section are clearly defined, the article is easily readable, and the topic is relevant to the readership. Limitations are adequately addressed. Conclusions are appropriate for the scope of the study. The paper is formally correct and it is clear its clinical relevance, and what this article should add to the body of knowledge on this topic.

### Thank you for your positive feedback.

---

## [Editor Report · Decision Letter 1]

23 Mar 2023

Use of Machine Learning to Identify Risk Factors for Coronary Artery Disease

PONE-D-22-32680R1

Dear Dr. Huang,

We’re pleased to inform you that your manuscript has been judged scientifically suitable for publication and will be formally accepted for publication once it meets all outstanding technical requirements.

Kind regards,

Donovan Anthony McGrowder, PhD., MA., MSc

Academic Editor

PLOS ONE

Additional Editor Comments (optional):

Dear Dr. Huang,

The manuscript entitled “Use of Machine Learning to Identify Risk Factors for Coronary Artery Disease” was revised in accordance with the reviewers’ comments and is provisionally accepted pending final checks for formatting and technical requirements.

Regards,

Dr. Donovan McGrowder (Academic Editor)

---

## [Editor Report · Acceptance letter]

4 Apr 2023

PONE-D-22-32680R1 

Use of Machine Learning to Identify Risk Factors for Coronary Artery Disease 

Dear Dr. Huang:

I'm pleased to inform you that your manuscript has been deemed suitable for publication in PLOS ONE. Congratulations! Your manuscript is now with our production department. 

Kind regards, 

on behalf of

Dr. Donovan Anthony McGrowder 

Academic Editor

PLOS ONE